# Boosting the Immune Response—Combining Local and Immune Therapy for Prostate Cancer Treatment

**DOI:** 10.3390/cells11182793

**Published:** 2022-09-07

**Authors:** Jakub Karwacki, Aleksander Kiełbik, Wojciech Szlasa, Natalia Sauer, Kamil Kowalczyk, Wojciech Krajewski, Jolanta Saczko, Julita Kulbacka, Tomasz Szydełko, Bartosz Małkiewicz

**Affiliations:** 1University Center of Excellence in Urology, Department of Minimally Invasive and Robotic Urology, Wrocław Medical University, 50-556 Wroclaw, Poland; 2Department of Molecular and Cellular Biology, Faculty of Pharmacy, Wroclaw Medical University, 50-556 Wrocław, Poland; 3Department of Drugs Form Technology, Faculty of Pharmacy, Wroclaw Medical University, 50-556 Wrocław, Poland

**Keywords:** metastatic castration-resistant prostate cancer, cancer vaccines, immunotherapy, focal therapy, combination immunotherapy, tumor immune microenvironment, in vivo vaccination

## Abstract

Due to its slow progression and susceptibility to radical forms of treatment, low-grade PC is associated with high overall survival (OS). With the clinical progression of PC, the therapy is becoming more complex. The immunosuppressive tumor microenvironment (TME) makes PC a difficult target for most immunotherapeutics. Its general immune resistance is established by e.g., immune evasion through Treg cells, synthesis of immunosuppressive mediators, and the defective expression of surface neoantigens. The success of sipuleucel-T in clinical trials initiated several other clinical studies that specifically target the immune escape of tumors and eliminate the immunosuppressive properties of the TME. In the settings of PC treatment, this can be commonly achieved with radiation therapy (RT). In addition, focal therapies usually applied for localized PC, such as high-intensity focused ultrasound (HIFU) therapy, cryotherapy, photodynamic therapy (PDT), and irreversible electroporation (IRE) were shown to boost the anti-cancer response. Nevertheless, the present guidelines restrict their application to the context of a clinical trial or a prospective cohort study. This review explains how RT and focal therapies enhance the immune response. We also provide data supporting the combination of RT and focal treatments with immune therapies.

## 1. Introduction

In 2020, prostate cancer (PC) was the second most frequent cancer and the fifth cause of cancer-related death among men. In more than half of the countries in the world, it was the most frequently diagnosed cancer in men [1]. While mortality rates are relatively low in comparison to other malignancies, metastatic castration-resistant prostate cancer (mCRPC) remains an incurable condition, with few treatment strategies providing any clinical benefit [2].

Focal therapies are minimally invasive treatment strategies used in the management of PC to provide local control of the disease, minimalizing the risk of possible complications. The first prospective clinical trials showed promising local disease control, especially with a short-term perspective [3]. The immunological impact of focal therapies, as well as the immunotherapy of PC itself, have been addressed by academic research for years. Thus, it was a substantial breakthrough when sipuleucel-T became the first therapeutic vaccine for patients with mCRPC approved by the United States Food and Drug Administration (FDA) and the first autologous cellular therapeutic vaccine in oncology [4,5]. Nevertheless, the clinical benefit of immunotherapy alone remains limited partially due to low-grade inflammation in the tumor microenvironment (TME), as well as uncontrollable cell divisions and the suppressed expression of tumor antigens [6,7]. However, both radiotherapy (RT) and various focal therapies have the potential to activate the anti-tumor immune response and, therefore, enhance the efficacy of immunotherapy [8,9,10].

The major purposes of this non-systematic review were to identify the immune effect of RT and focal therapies, including high-intensity focused ultrasound (HIFU), cryotherapy, photodynamic therapy (PDT), and irreversible electroporation (IRE), and to compile available knowledge on different combinational therapies including both a focal and an immunotherapeutic component. To provide background data on the immunology of PC, we thoroughly described the TME and various immune evasion mechanisms. Furthermore, for better understanding of the current status of immunotherapy, we summarized the most recent scientific data on different management options in PC, including immune checkpoint inhibitors (ICIs) and cancer vaccines.

## 2. Data Acquisition

For the purposes of this non-systematic review, we conducted a comprehensive English language literature search for original articles, meta-analyses and reviews using PubMed and grey literature through June and July 2022. We searched for various combinations of the following terms: prostate cancer; immunotherapy; focal therapy; cancer vaccine; combination therapy; tumor microenvironment. We found 1669 related articles, and the final number of papers selected for this manuscript was 392. Studies with the highest level of evidence and relevance to the discussed topics (275) were selected, with the consensus of the authors.

## 3. Immunological Background of Prostate Cancer

### 3.1. Prostate Cancer Microenvironment

The microenvironment of PC consists of numerous elements, including both neoplastic cells and diverse host cells. The host component comprises stromal cells, the extracellular matrix, endothelial and vascular cells, immune cells, and various soluble factors, such as IL-6 and receptor activator of nuclear factor kappa-B ligand (RANKL) [11]. The TME in PC plays an ambiguous role in carcinogenesis. Particularly, the impact of the immune system is highly complex, as both innate and adaptive immune response mechanisms can provide anti-neoplastic activity, as well as propagate carcinogenesis [12]. For example, cytotoxic T lymphocytes (CTLs), one of the most important cancer cell killers, secrete transforming growth factor-beta (TGF-β), which both supports tumor growth and induces immune suppression [13].

There is a multitude of mechanisms affecting the TME in PC and contributing to immunosuppression, including the inhibition of neoantigen expression, instability of rapid cell division, DNA damage response (DDR) gene defects, decreased human leucocyte antigen (HLA) expression, phosphatase and tensin homolog (PTEN) protein loss, and dysfunction of interferon (IFN) type I signaling [7].

### 3.2. T Cell Infiltration

Many immune cell types play a role in TME functionality, although the T cell population, especially CTLs, are considered most vital [14]. They are the key elements of the physiological cancer immunity cycle, which is briefly summarized in Figure 1 [15]. T cells are recruited from peripheral blood after antigen-presenting cells (APCs), specifically dendritic cells (DCs) and macrophages, capture neoantigens released by the tumor. Presenting the abovementioned antigens to CTLs using a major histocompatibility complex (MHC) is called priming and takes place in the local lymph nodes. This results in recruiting and stimulating more T cells, including CD4^+^ cells. CTLs infiltrate the tumor, recognize cancer cells, and kill them. Neoantigens are then released and the process comes full circle [15]. The localization and density of tumor infiltrating lymphocytes (TILs) and memory T cells within the center of the tumor and its margins were the foundation for creating the “immunoscore”. It divides tumors into two groups: T cell inflamed (“hot”) and non-T cell inflamed (“cold”) [16]. This immune contexture is significant in the efficacy of therapy in a variety of cancers. Many publications indicate that a high level of TILs shows a positive prognostic value [17,18,19,20,21,22]. PC is primarily described as a “cold” tumor, with a low inflammation burden and immune activation [23]. However, the impact of the TME on PC oncological outcomes is unclear [7]. Some studies show that the high intratumoral density of CTLs is associated with improved cancer-specific survival (CSS) in PC patients undergoing RP [24,25]. Others show that the higher the level of CTL infiltration in PC, the greater the risk of distant metastases and biochemical recurrence [12,26]. Although the connection between inflammation and tumorigenesis remains unclear, one of the main goals of various local pre-immunotherapy techniques is to propagate the inflammation of the TME, converting it to inflamed and susceptible to immunotherapy [27].

### 3.3. Regulation of the T Cell Response

After a T cell is initially activated during priming, the second step of activation takes place: binding of the costimulatory molecules, CD80 (B7-1) or CD86 (B7-2), which serve as ligands on APCs, and CD28, a receptor expressed on T cells [28,29,30]. The cytotoxic T lymphocyte antigen 4 (CTLA-4 or CD152) is a co-inhibitory glycoprotein receptor expressed on the surface of the T cell, competing with CD28 for B7 ligands. CTLA-4 is induced after T cell activation (except for regulatory T cells (Tregs), which continuously express it), and due to its higher affinity for B7 molecules, it successfully outcompetes the CD28 receptor [31,32,33,34]. The B7:CTLA4 interaction leads to inhibition of the cell cycle progression through IL-2 accumulation [35,36]. Programmed death receptor 1 (PD-1) is another co-inhibitory receptor on the surface of T- and B cells. PD-1 ligand 1 (PD-L1 or CD274) and PD-1 ligand 2 (PD-L2 or CD273) are two known ligands for the PD-1 receptor, expressed on macrophages, DCs, and other immune cells [37]. Although the interaction of PD-L2 and PD-1 also has an immunosuppressive outcome, it is the PD-L1:PD-1 binding that induces the conversion of naïve T cells into Tregs [38,39,40,41]. CTLA-4, PD-1, and its ligands are parts of the B7 superfamily of molecules and are the most vital immune checkpoints (ICPs) [42].

### 3.4. Immune Evasion Mechanisms

Cancer cells have developed several immune evasion mechanisms associated with TME components. Immune evasion may be described as the entirety of biochemical interactions leading to the suppression of the natural immune response to tumor cells. The spectrum of possible “back doors” can be generally divided into a few mechanisms. These include: (1) immune evasion through immune cells (most notably Tregs), (2) synthesis of immune-suppressive mediators, and (3) defective expression of surface neoantigens [43]. Additionally, important contributors to the TME, affecting cancer progression and the response to therapy, are hypoxia and glucose restriction, which have already been described in a multitude of malignancies [44,45,46].

#### 3.4.1. The Role of Specific Immune Cells

One of the cancer immune evasion mechanisms is CD4+ CD25+ FOXP3+ Tregs activity, as its physiological role is to modulate effector T cells to support immunological tolerance to self-antigens (self-Ags) [47,48,49]. Tregs drawn by the tumor have higher suppressive properties compared to circulating Tregs and are able to inhibit the anti-tumor activity of other immune cells directly using cell–cell interactions or indirectly through synthesis and the secretion of mediators, e.g., TGF-β, interleukin 10 (IL-10) [50,51]. Many tumor-associated Ags are expressed by host cells and can therefore act like self-Ags, which further emphasizes the role of Tregs in immune evasion [52,53].

Myeloid-derived suppressor cells (MDSCs) are another heterogeneous group comprising immature DCs, granulocytes, and macrophages. Overproduction and concentration of these cell types in an inflammatory environment are correlated with the immunosuppressive qualities of the TME [54,55]. Their functions include the inhibition of CLTs through various mechanisms (e.g., producing reactive oxygen species (ROS) or interactions with the T cell receptor [TCR]), suppressing natural killer (NK) cells, and Tregs induction [56,57,58,59,60]. The level of MDSCs correlates with the stage of PC and applied treatment, as well as with the serum levels of crucial inflammatory mediators—IL-6 and IL-8 [61,62,63].

DCs are the most efficient APCs, but their functionality is mutilated due to the modulatory activity of tumors. Impaired DCs have lower levels of CD80, CD86, and CD40, thus they cannot present antigens and activate T cells effectively enough [43,64,65]. The role of CD40 is highly complex as it connects the T- and B cell responses. Namely, when DCs remain active and secrete IL-12, they may interact with CD40L on both T cells and B cells [66,67,68,69]. The first interaction induces the Th1 and IFN-γ secretion by the T cells, and the second induces the class switching between IgG and IgA in B cells [70,71,72]. In addition, the reciprocal expression of CD40 and CD40L on DCs, T cells, and B cells links the humoral and cellular immune response, thus the reduced level of CD40 might lead to impairments in both responses [72].

Tumor-associated macrophages (TAMs) are another important group contributing to the PC TME. TAMs, especially the M2 type, can stimulate tumor growth through the secretion of various mediators such as TGF-β, IL-10, and vascular endothelial growth factor (VEGF) [43,73,74,75]. Overexpression of TAMs in PC is correlated with unfavorable oncological outcomes in patients with PC, including biochemical recurrence (BCR), or worse, distant metastasis-free survival [76,77,78].

Furthermore, increased concentration of TAMs is observable in metastatic lymph nodes in comparison to the primary prostate tumor [79,80]. Shortage of available data on other TME differences in primary and metastatic PC prevents us from presenting a thorough comparison, but it is worth mentioning that this immunological heterogeneity has already been described in other malignancies, including breast, lung, colorectal cancers, and brain tumors [81,82,83,84,85,86,87].

The role of B cells in immune evasion is not well understood in the case of PC. However, B cell infiltration has prognostic significance in different cancers such as breast cancer and melanoma [88]. B cell TILs secrete a significant member of the TNF family, lymphotoxin (LT), which promotes survival and proliferation of androgen-deprived cells, therefore encouraging castration-resistant PC (CRPC) development [89].

#### 3.4.2. Immunosuppressive Mediators

There are many immunosuppressive cytokines which aid tumors in immune evasion through the promotion of tumor proliferation, chemoresistance, angiogenesis, and migration, and these are most notably TGF-β, VEGF, IL-6, RANKL, and the CXCL family [90]. TGF-β is one of the most vital mediators, acting both as a direct growth-promoting factor, as well as a stimulator of CD4+ T cells–Tregs transformation [91,92,93]. Its other roles include promoting angiogenesis, downregulating HLA-1 expression, and inducing epithelial–mesenchymal transition (EMT) [90,94,95,96,97]. Another important cytokine is VEGF, which also contributes to tumor growth, as well as inhibiting DC differentiation. A similar role is performed by cancer-associated ganglioside antigens, which conduct an immunosuppressive activity through impairing CTLs and DCs [43,98,99].

#### 3.4.3. Dysfunctional Expression of Surface Neoantigens

MHC Class I proteins are found on nucleated cells and platelet surfaces, and their role is to be recognized by CD8+ T cells, which trigger the immune response against certain antigens by activating T cells and leading to target cell destruction [5,100,101]. Decreased MHC I presentation of tumor-associated antigens is one of the immune evasion mechanisms of PC [102,103]. A study by Yitalo et al. revealed heterogeneity in MHC I expression in PC. Bone metastases showed noticeably lower MHC I expression in comparison to the primary tumor. Moreover, a subgroup of metastases showed high MHC I expression, while the majority of metastases showed a decrease. These findings indicate that there are molecular subtypes of PC which may be less responsive to immunotherapy [104].

#### 3.4.4. Monitoring of the Immune Response

The immune effects provided by immunotherapy or initiated by focal therapies are difficult to unequivocally assess. There is no gold standard of immunological activity, and various studies focus on different aspects of induced immunomodulation. Some of the most valid indicators of this state include: immunogenic cell death; concentration of CTLs, DCs, and other immune cells; the expression level of certain antigens and factors (e.g., PD-L1, MHC, IL-1, IL-6, interferons); and the abscopal effect [105,106,107]. Nevertheless, a well-designed randomized clinical trial comparing the immune effects of the abovementioned treatment strategies would be vital to adequately evaluate the immunomodulatory effects provided by focal therapies.

## 4. Immunotherapeutic Options for Prostate Cancer

### 4.1. Immune Checkpoint Inhibitors

ICIs are novel treatment options gaining more and more interest as they already appear to act as successful strategies in cancers such as melanoma and lung cancer [108,109,110,111,112,113,114]. Among CTLA-4 inhibitors used in oncology, there are ipilimumab and tremelimumab, while the most pivotal PD-1 inhibitors comprise nivolumab and pembrolizumab; atezolizumab, avelumab, and durvalumab belong to PD-L1 inhibitors [115,116,117].

There have been a few major studies on ICI efficacy in PC management in recent years; most trials investigated various immunotherapy combinations, especially with other immunotherapeutics or with systemic chemotherapy, or ADT. The preliminary results of the study by Sharma et al. (CheckMate 650, NCT02985957) showed promising anti-tumor activity of the nivolumab and ipilimumab combination in patients with mCRPC. The authors reported 25% objective response rate (ORR) in the pre-chemotherapy arm, and 10% ORR in the post-chemotherapy arm, as well as 5.5 and 3.8 months median radiographic progression-free survival (PFS), and 19.0 and 15.2 months median OS. However, research expansion is necessary to further assess the clinical outcomes of this therapy [118]. In the randomized phase III trial by Powles et al. (The IMbassador250 trial, NCT03016312), the combination of atezolizumab and enzalutamide failed to meet the endpoint of improved OS. However, the authors pointed out that proper patient selection may be vital to identify subgroups of patients who are susceptible to this therapy [119]. The phase III randomized study by Agarwal et al. (CONTACT-02, NCT04446117) will evaluate the combination of atezolizumab and cabozantinib vs. second novel hormone therapy (abiraterone or enzalumatide) in patients with mCRPC [120].

The European Association of Urology (EAU) guidelines indicate that pembrolizumab may be a valuable additional management strategy for mCRPC patients with high microsatellite instability [121,122]. This further suggests that the increase in immunotherapy efficacy in PC may require meticulous patient selection.

Immune checkpoint blockade in PC remains a poor monotherapeutic tool [123,124]. Among the reasons for this is the low level of T cell infiltration, the “cold” immunogenic profile of the tumor, mutational burden, and immune evasion mechanisms [125,126,127]. Thus, the interest in boosting the immune response, using other therapies beforehand, is now rapidly growing. It is believed that various therapy combinations can subdue tumor resistance to ICI therapy [128]. Studies evaluating ICI combinations with each other and with focal therapies show more promising results in comparison to monotherapeutic approaches [118,123]. However, the study by Palmer et al. revealed that the combination of ICIs with various other strategies (e.g., other immunotherapy, chemotherapy) shows no enhancement (additivity or synergy) in comparison to ICIs alone in 13 phase III clinical trials [129].

For the purpose of this review, we present results of various studies on the combination of ICIs and focal therapies in the sections concerning specific local treatment strategies.

### 4.2. Cancer Vaccines

Cancer vaccines usually consist of specific neoantigens and adjuvants that boost the immune response against these antigens. Their role is to support the adaptive immune system of the patient in neutralizing cancer cells [130]. Cancer vaccinations have already been combined with RT and focal therapies, due to their immunomodulatory effect, in various clinical trials, evaluating their efficacy in numerous malignancies including PC, melanoma, pancreatic, and lung cancers [131,132,133,134,135,136,137,138,139,140,141].

#### 4.2.1. Dendritic Cell Vaccines—Sipuleucel-T, DCvac/PCa, and Others

As mentioned above, DCs are one of the most important features of the immune system; they are the most efficient APCs, which are not only able to activate T cells (both Tregs and CTLs) but also NK cells. DC vaccines require blood-derived DCs, pulsing them ex vivo with the tumor-associated antigen, activating them using the specific adjuvant, and then reinjecting them into the patient [142]. The first DC vaccine approved by the U.S. Food and Drug Administration (FDA) was sipuleucel-T (Provenge^®^), and so far, it remains the only DC vaccine for mCRPC [143,144]. Sipuleucel-T promotes an immune response against tumor cells using prostatic acid phosphatase (PAP) antigen-activated DCs [42]. A double-blind, placebo-controlled, multicenter phase III trial compared this DC vaccine to the placebo group, with the results of a 22% reduction in the risk of death and more than 4 months of improvement in overall survival (OS) [4]. Another trial showed even greater improvement in OS (up to 8.1 months), if sipuleucel-T therapy is extended by APC8015F, a variant of the DC vaccine prepared from cryopreserved cells which were frozen for future use [145].

DC vaccines are a very promising therapeutic tool, although they require further clinical trials and more attempts to combine them using different approaches [146]. There is only one ongoing trial assessing the combination of sipuleucel-T and other therapies: sipuleucel-T plus stereotactic ablative body radiation (SABR) (NCT01818986, phase II). A different phase III trial evaluates the efficiency of sipuleucel-T in reducing the progression of CRPC. The study includes active surveillance patients (the ProVent Study; NCT03686683).

DCvac/PCa is an autologous DC-based vaccine, in which DCs are pulsed with killed lymph node carcinoma of prostate (LNCaP) cells. Several clinical trials have investigated its efficacy in PC. However, only a few of them examine its combinations with focal therapies or RT. Fucikova et al. assessed the DCvac/PCa impact on PSA in patients with rising PSA after RP or salvage RT. PSA doubling time was significantly elongated in this variant [147]. Although the DCvac/PCa immunological impact is quite well documented, its translation to clinical benefits is needed, and further clinical trials are required, especially concerning different combinations of therapies. A recent clinical phase III trial (the Viable) by Vogelzang et al. investigated the DCvac/PCa combination with docetaxel and prednisone. The therapy failed to improve OS in patients with mCRPC [148].

Other DC-based vaccines that have been tested in the last decade in PC patients are prostate-specific membrane antigen (PSMA) and the survivin-loaded DC vaccine, mucin 1 (MUC1) vaccine, and the T cell receptor ɣ alternate reading frame protein (TARP) vaccine [149,150,151,152].

#### 4.2.2. PROSTVAC—A PSA-Based Viral Vector Vaccine

One of the trailblazing PC vaccines is PROSTVAC (PSA-TRICOM), which comprises two recombinant poxvirus vectors containing transgenes for PSA and three costimulatory molecules: B7.1, ICAM-1, and LFA-3 [153,154]. A phase II trial analyzing neoadjuvant PROSTVAC in patients awaiting RP showed an increase in CD4+ and CD8+ T cell infiltration of the tumor, as well as a peripheral immune response to neoantigens in 13 out of the 25 patients [155]. However, this promising immune response does not yet translate into a clinical advantage. In a phase III trial, Gulley et al. concluded that despite the fact that the therapy was well-tolerated and safe for patients, treatment disappointingly showed no impact on median OS and alive without event (AWE) in patients with mCRPC [156]. Parsons et al. evaluated the preventive value of PROSTVAC in patients with localized PC which is managed by an active surveillance strategy. Although some initial data on the immunological effect of the vaccine are already available, we are looking forward to the summary of this phase II trial in the future (NCT02326805) [157].

Several ongoing clinical trials are investigating different combinational management strategies including PROSTVAC. These are evaluating, among others, the combination with nivolumab (NCT02933255, phase I/II) or nivolumab and ipilimumab (NCT03532217, phase I), with CV301 (a poxviral vaccine) and M7824 (a protein targeting PD-L1 and TGF-β) (NCT03315871, phase II), docetaxel (NCT02649855, phase II), or enzalutamide (NCT01867333, phase II).

TroVax is another viral vector, 5T4 (oncofetal glycoprotein) targeting vaccination. It is characterized by a good immune response in mCRPC and the potential to efficiently combine with docetaxel [158,159].

#### 4.2.3. Peptide-Based Vaccines

Among peptide-based vaccines, one of the most interesting is GX301, consisting of four telomerase peptides and two adjuvants—Montanide ISA-51 and Imiquimod. Fenoglio et al. assessed its potential in a phase I/II clinical trial, revealing its immunological response in PC and renal cell cancer (RCC). An increase in PFS and OS were also observed [160]. Filaci et al. evaluated GX301 efficiency and immunological impact in mCRPC. The therapy did not increase OS, although they observed that a higher number of drug administrations was correlated with an increased immunological response [161].

Cell division-associated 1 (CDCA1) peptide vaccination was a topic of research in a phase I clinical trial by Obara et al. CDCA1 is a peptide overexpressed in a few malignancies, including PC. The authors indicated that the vaccine is well-tolerated, and it boosts the immunological response in patients with CRPC [162,163].

Other peptide-based vaccinations include the personalized peptide vaccination (PPV), which includes the administration of different HLA-matched peptides, multi-peptide vaccines, and a vaccine targeting the Ras homolog gene family member C (RhoC vaccine, RV 001, or onilcamotide) [164,165,166]. However, a recent phase IIb study of RhoVac (RV 001) failed to show its superiority over the placebo in PC (BRaVac, NCT04114825). Further phase II and III trials investigating various peptide-based vaccinations are required in the future.

#### 4.2.4. Whole Tumor Cell Vaccines

GVAX is a vaccine consisting of genetically modified PC cells, which have undergone radiation. Studies suggest that this vaccination induces immune response through the activation of DCs and MDSCs [167]. A combinational therapy with ipilimumab has been investigated in a phase I trial by van den Eertwegh et al., which showed that GVAX is well-tolerated and safe for patients with mCRPC [168]. Once again, further clinical trials are required [169].

## 5. Focal Ablation and Immune Therapy Combination

### 5.1. High-Intensity Focused Ultrasound

Recently, HIFU appeared as a potential neoadjuvant-like therapy, serving as the first step of immunotherapeutic treatment. HIFU itself has already made an appearance in guidelines concerning PC treatment options, although only as an investigational therapeutic tool or as salvage therapy [170]. Essentially, this focal management strategy uses highly focused ultrasound waves to ablate cancer cells and has been investigated extensively in PC patients in a multitude of clinical trials since the late 1990s [171,172,173,174,175,176,177,178,179,180,181]. The most important benefit of HIFU is that it is minimally invasive when compared to surgical treatment, and it is devoid of systemic toxicity in comparison with androgen deprivation therapy (ADT) or chemotherapy. Nevertheless, possible adverse effects may quite frequently occur, and they include erectile dysfunction, urinary tract infections, rectal injuries, and more [182,183]. The properties of HIFU can be divided into a few groups—ablative and non-ablative (mechanical), immune, and biological effects. Induced activity depends on a multitude of factors including frequency, pressure, duty cycle, treatment time, achieved temperature, tissue susceptibility, and more. This allows us to distinguish several possible technique variants, such as thermal ablation, thermal stress and hyperthermia, mechanical perturbation, and histotripsy [184]. However, the first and foremost effect of HIFU is thermal ablation (by heating tumor tissue above approximately 55 °C), resulting in coagulative necrosis, combined with additional cavitation formation. The most interesting secondary effect is anti-tumor immunity induction [185,186].

The immunotherapeutic effect of HIFU has recently been investigated in many kinds of malignancies. Hu et al. confirmed HIFU promotes DC infiltration and activation in mice bearing colon adenocarcinoma and indicated that the mechanical components of this procedure may be successfully combined with other types of therapy [187]. Ran et al. showed that HIFU increases peripheral blood CD3+, CD4+ levels and the CD4+/CD8+ ratio, enhances CTL cytotoxicity against murine hepatocarcinoma, and inhibits tumor growth and progression in mice [188]. The impact on the CD4+/CD8+ ratio has been observed in the past by Rosberger et al. [189]. The activation of anti-tumor immunity promoted by HIFU can be partially explained by tumor debris “left-over” antigen immunogenicity, which was demonstrated by Zhang et al. in the murine hepatocellular carcinoma model [190]. Similar investigations have been conducted with other malignancies, such as melanoma, neuroblastoma, and pancreatic cancer [191,192,193,194,195,196]. Wu et al. researched tumor debris immunogenic properties in 23 patients with breast cancer. Using HIFU, they ablated primary tumors and evaluated the expression of tumor antigens and heat-shock protein 70 (HSP-70), while also pointing out the immunogenic potential of neoplastic debris [197].

Sonodynamic therapy (SDT) is another promising strategy concerning the usage of ultrasound. It is based on the application of sonosensitizers, which is followed by their activation with ultrasound. Activated particles then transfer the energy to oxygen accumulated in the TME, creating ROS which kill or damage tumor cells [198,199]. HIFU and the spectrum of ultrasound-based therapies in general are still very modern approaches used for enhancing immune response. Further investigation is required, especially concerning PC.

### 5.2. Cryotherapy

Cryoablation or cryotherapy performed either as a focal therapy, or as the whole-gland procedure, is an ablation technique using extremely low temperatures to induce both necrosis and apoptosis of tumor cells [200,201,202]. With the use of special cryoprobes, liquid nitrogen or argon, passing from high pressure to an atmospheric pressure revealing its cooling effect, is implemented inside a prostate gland. Although it may be used as a monotherapy, for this review, we will only focus on its immunomodulatory activity and its combination with immunotherapy.

Cryotherapy has great potential to enhance the immune response, due to its significant preservation of tumor antigens and cytokines, compared to other ablation techniques based on high temperatures and rather than hypothermia [203]. It is believed to leave the intracellular molecules of tumors intact and, through attracting the immune system using these factors, stimulate tumor-specific immunity. However, cryotherapy can prompt both immunostimulatory and immunosuppressive responses, which are strongly dependent on the type of induced cell death; studies suggest that necrosis, occurring mainly in the inner zone of the tissue, causes tumor cells to release danger-associated molecular patterns, which boost the immune response through the maturation of DCs, and consequently, T cell activation. However, apoptosis primarily occurring in the peripheral margin of the ablated organ leads to a lack of secretion of danger signals, therefore contributing to immunosuppression [204,205,206]. The cryoimmunological effect is further described by the term “abscopal effect”. This rare phenomenon refers to the systemic immunological impact a focal therapy has and primarily refers to the reduction in a metastasis preceded by a localized treatment in a different location [207]. This process was proved to be mediated by CD8+ T cells and correlated with a low level of CD4+CD5+ Tregs, as well as an increased level of IFN-ɣ [208,209].

Various investigations have been conducted on the theme of cryoimmunological therapy, both in murine models and in clinical trials. For instance, Gaitanis and Bassukas researched the impact of immunocryosurgery on basal cell carcinomas (BCC). Their study indicated that cryoablation combined with the TLR7 agonist, imiquimod, can be a very effective substitute for surgical treatment for BCC under 20 mm in diameter [210]. In another study, Lin et al. prospectively evaluated allogeneic NK cell immunotherapy combined with cryosurgery in renal cell carcinoma (RCC). They once again proved an additive effect of the two therapies [211]. The same group of researchers conducted similar investigations in patients with lung and hepatocellular cancers, with similarly favorable results [212,213].

So far clinical trials including the combination of cryosurgery and immunotherapy in patients with PC are rarely conducted. One of them is a therapy using granulocyte–macrophage colony-stimulating factor (GM-CSF), a cytokine regulating the function of granulocytes and macrophages, as well as promoting the survival of DCs [214,215]. These investigations revealed that GM-CSF administration enhances INF-ɣ secretion by T cells on the base of prior cryoablation procedures, as well as the fact that GM-CSF increases levels of prostate-specific and nonspecific antigens. Ross et al. examined cryosurgery combined with short term ADT and pembrolizumab, a PD-1 inhibitor, proving local disease control but questioning its potential for the management of systemic disease [216].

### 5.3. Photodynamic Therapy

PDT is an example of another targeted treatment option, that has already been used as an alternative to radical therapies, with the intention of reducing levels of side effects, while maintaining favorable oncological outcomes [217]. This focal therapy is based on the usage of a laser of a specific wavelength, which activates the photosensitizer (PS), administered systemically or locally, and therefore generates ROS resulting in necrosis of the tumor cells [218]. Depending on the qualities of photoagents, different effects can be achieved. Photothermal therapy (PTT) is a subtype of phototherapy that is different from PDT as it engages PS properties not to produce ROS but to execute a thermal effect through the conversion of absorbed laser light into heat [219].

As for PDT in PC, researchers point out the high efficacy and low level of adverse effects of vascular-targeted photodynamic therapy (VTP) in comparison to other therapies, while addressing the great need for long-term benefit evaluation in randomized clinical trials (RCT) [220]. Rastinehad et al. introduced the results of a clinical trial in which they used gold-silica nanoshells (AuroShells) to conduct PTT in 15 patients with PC. The study revealed high-profile feasibility of the procedure, and once again pointed out its low-rate adverse effects burden [221]. Another study by Azzouzi et al. compared padeliporfin VTP with an active surveillance strategy in a phase III RCT. They evaluated VTP as a safe and effective treatment for low-risk, localized PC, with a longer time to progression and a higher proportion of negative biopsy results in comparison to active surveillance [222]. On the other hand, a review of this investigation, initiated by the Oncologic Drugs Advisory Committee within the FDA, resulted in voting against approval of this therapeutic strategy in the United States [223]. There were many reasons for this, including, above all, an unfavorable benefit–risk profile of this therapy, postulated by the panelists [224]. Nevertheless, more clinical trials are required, evaluating different doses of various PS, varying laser wavelengths, and manipulating other parameters [223].

However, more and more papers recently have been turning their attention to the immunological aspects of PDT as it propagates inflammatory response, induces necrosis, and promotes recruitment of neutrophils and other immune cells [225,226,227,228,229]. Furthermore, PDT can promote immune cells and engage them to eradicate distant metastases [230]. Therefore, the term photoimmunotherapy (PIT) has been forged, and it may be described as a combination of immunogenic properties of PDT and immunotherapy treatment [231]. The immunological effect obtained by PDT is complex and multi-level. First of all, it directly affects immune cells through the recruitment of neutrophils, DC maturation, and macrophage activation, as well as the accumulation of CTLs and affecting them through regulation of NK cell migration [225,232]. Secretion of IL-1 α/β, IL-6, IL-8, IL-10, and IL-12 is boosted as is the release of a few secondary inflammatory mediators, including thromboxane and prostaglandins [233]. Furthermore, a few strategies concerning the combination of PDT with different immunotherapeutic strategies have been conducted, and their results are promising. Li et al. evaluated the synergistic effect of CTLA-4 antibodies and single-walled carbon nanotube-glycated chitosan complex (SWNT-GC), which acts primarily as a light absorber, in metastatic mammary tumors in mice. Local administration was then followed by PTT. The results showed that this strategy prolonged survival time, suppressed primary tumors, and inhibited metastases [234]. Huang et al. introduced a drug conjugate consisting of protoporphyrin IX and NLG919, a potent indoleamine-2,3-dioxygenase (IDO) inhibitor, which is applied to the cells through liposomal delivery (PpIX-NLG@Lipo). They showed its strong ability to generate ROS after the phototherapeutic procedure, as well as its potential to increase CD8+ T cell infiltration [235]. Kim et al. investigated the impact of PDT with a Ce6-embedded nanophotosensitizer (FIC-PDT) with ripasudil, a rho-kinase (ROCK) inhibitor, on the immune response in mice with uveal melanoma. Their research indicated that this combination demonstrates a vaccine-like function, realized by evoking immunogenic cell death and stimulating APCs, which leads to CD8+ T cell priming and their accumulation in the primary tumor, and, in further synergy with the anti-PD-L1 antibody, to metastasis inhibition [133].

Nagaya et al. presented the effects of near-infrared photoimmunotherapy (NIR-PIT) with a prostate-specific membrane antigen (PSMA) antibody in the PC cell line. The anti-PSMA antibody was conjugated to the light-absorbing agent, IR700DX. This antibody–PS conglomerate was observed to bind cell-specifically and to effectively kill PC cells after activation using NIR-PIT, with over two-thirds of the investigated tumors cured [236]. Research on the same topic was conducted by Watanabe et al., and it pointed to the possibility of using only fragments of anti-PSMA antibodies instead of the full antibodies, which may clinically translate to a more thorough penetration of the tumor milieu. Using smaller parts of antibodies should also shorten the time gap between the injection of the PS and NIR-PIT [237].

### 5.4. Irreversible Electroporation

IRE is the permeabilization of cell membranes with electrical pulses, which affect membranous electrochemical potentials, creating pores in a lipid bilayer [238]. IRE has been already used in PC management, both as a focal therapy, and as the whole gland ablation [239,240,241,242,243,244,245,246]. The procedure is based on needle electrodes, which are placed inside or nearby the targeted tissue. Then, short electrical pulses are delivered, which induces cell death through a non-thermal mechanism [247]. Despite its role in the immune response is still unexplored, IRE seems to have immunomodulatory activity. The most pivotal immunological effect of IRE is a decrease in Tregs in the TME; additionally, a decrease in MDSCs occurs as well [248].

The field of IRE–immunotherapy combinations in treating malignancies is still uninvestigated, although there are a few articles, especially on pancreatic cancer. Yang et al., for example, revealed a connection between IRE and tumor-associated immune evasion in a mice model of pancreatic ductal adenocarcinoma (PDAC). They indicated that IRE combined with the DC cancer vaccination increases the level of tumor-infiltrating cells including CD8+ T cells and granzyme B+ cells in PDAC [249]. Similar investigations have been conducted by Zhao et al. and by He et al. Both studies showed promising results from the combination of IRE and PD-1 inhibitors in mice with PDAC [250,251].

A study by Burbach et al. examined the combination of IRE and ICI in mice with PC. Focal treatment using IRE combined with ICI led to the expansion of tumor-specific CD8+ T cells in blood and the TME [252].

## 6. Radiation and Immune Therapy Combination

RT has been used as a management strategy both in PC and in many other malignancies for years. Its primary property, exploited for tumor treatment purposes, is the effect on double-strand DNA, leading to its breakdown, and thus resulting in cell death, majorly through senescence and slightly less frequently through mitotic catastrophe, apoptosis, and necrosis [253]. Traditionally, RT was considered to be a therapy of immunosuppressive qualities, therefore its combination with immunotherapy appeared to be irrational at first [254]. However, rapidly growing interest in the TME affected the way RT is perceived, as its relationship with the immune system is far more complex and ambiguous [255,256].

### 6.1. Immunomodulatory Effect of RT

An immune-stimulating effect of RT is generally achieved through induced cell death and modulating the composition of the TME [257]. One of the initial steps following tumor cell damage is the enhanced release of damage-associated molecules, such as calreticulin, adenosine triphosphate (ATP), GM-CSF, high-mobility group box 1 (HMGB1), and heat shock proteins (HSPs) [258,259]. Afterwards, these damage signals activate DCs and APCs, which takes place in lymph nodes and lead to priming naïve T cells as a consequence [259]. Additionally, one of the radiation effects is the release of other inflammatory molecules, such as chemokines (e.g., CXCL10 and CXCL16) and other cytokines, including IL-1β, TNF-α, and type 1 and 2 interferons, which further contribute to the increase in inflammation in the TME [260]. Finally, RT triggers upregulation of MHC I, the NKG2D ligand, Fas/CD95, and other co-stimulatory molecules, resulting in cell death and further antigen exposure [259,260].

Eckert et al. investigated the impact of RT on the immune system in 18 patients with localized PC. The study revealed the ambiguous effect of ionizing radiation in RT resulted in a decrease in absolute leukocyte and lymphocyte counts and an increase in Tregs and NK cells counts over eight weeks after radiation. However, during RT, an increase was observed in all immune cells counts excluding Tregs. Importantly, the percentage of CD8+ T cells showed its peak early during RT [261]. Nevertheless, Harris et al. researched a combination of RT and immunotherapy in a transgenic murine model and observed that the anti-tumor immune response occurred when immune therapy was administered 3 to 5 weeks after RT [20]. This further suggests the existence of a certain type of therapeutic time window, in which the immunostimulatory properties of RT are emphasized, and the immunosuppressive component is partially inhibited. Nickols et al. researched the impact of stereotactic body radiotherapy (SBRT) on immunological homeostasis in a clinical trial evaluating the resected prostate specimens of 16 patients. While prostates without SBRT were mainly lymphoid diverse, specimens after SBRT were immunologically dominated by myeloid cells [262]. Keam et al. proved in their 24 patient clinical trial that high dose-rate brachytherapy (HDRBT) has substantial potential in enhancing inflammation in the prostate. In response to HDRBT, an increase in CD4+ T cells, macrophages, and DCs counts was observed. Moreover, they evaluated the tumor inflammation signature (TIS) and concluded that 80% of immunologically “cold” tumors were converted to “intermediate” or “hot” types [27].

Interestingly, RT is another management strategy with proven abscopal effect, hence resulting in the regression of metastases, probably due to the outburst of tumor-associated antigens. This extremely rare effect is observed more often when RT is combined with immune therapy, particularly with checkpoint inhibitors [258]. Dudzinski et al. studied the combination of anti-PD-1 or anti-PD-L1 and radiation in mice, and they not only observed an increase in median survival in comparison to the drug alone (70% longer for anti-PD-1 and 130% for anti-PD-L1) but also detected the abscopal effect—a regression of unirradiated distant metastases [263].

### 6.2. Radioimmunotherapy in Murine Models

In the research concerning the effects of RT and immunotherapy combination in mice, there have been a few distinguishing articles, including the paper from Wada et al. They assessed the efficacy of this therapy (the immunotherapeutic component was GM-CSF) using an autochthonous model of PC. Improved OS and increase in the effector-to-regulatory TILs ratio, as well as treatment effect in both the primary tumor and metastases, were observed [264]. Another investigation by Philippou et al. assayed the combination of anti-PD-L1 and RT and its impact on the TME in PC. They observed macrophage and DC counts increase, as well as an upregulation of PD-1/PD-L1 in both arms of the study 7 days after RT. Radiation was observed to delay tumor growth and affect the TME immunological composition. However, PD-L1 inhibition administered in one of the arms did not affect tumor growth delay when compared to monotherapy [265]. Table 1 presents ongoing trials evaluating different combinations of RT and immunotherapy in PC management.

### 6.3. Radioimmunotherapy in Prostate Cancer

The efficiency of radioimmunotherapy in patients with PC has been willingly explored in clinical trials for the last 10 years. Slovin et al. assessed the combination of anti-CTLA-4 antibody, ipilimumab, with external beam radiotherapy (EBRT) in comparison to the drug alone. This phase I/II study on 50 patients evaluated adverse effects, defining them as manageable and indicating anti-tumor activity [266]. In another investigation, a phase III trial concerning ipilimumab versus placebo after radiotherapy in patients with mCRPC that progressed after docetaxel chemotherapy was conducted by Kwon et al. No notable difference in OS was found, although ipilimumab use was associated with a decrease in PSA levels and an increase in PFS. Additionally, an OS increase was observed in the ipilimumab subgroup without visceral metastases, with non-raised or mildly raised alkaline phosphatase and without anemia. Accordingly, the authors suggested that a specific constellation of prognostic features could potentially enhance the clinical outcomes of radioimmunotherapy [267]. The final analysis of this phase III trial revealed that OS was two to three times higher at 3 years and beyond in favor of the radiotherapy and ipilimumab combination [267]. Different clinical trials assessing nivolumab and brachytherapy or EBRT, as well as sipuleucel-T and EBRT combinations indicated that these therapies are safe and well-tolerated. The immunogenic effect and anti-tumor activity of radiation with nivolumab were observed, while radiation with sipuleucel-T showed no particular increase in the immune response [268,269]. Another phase II trial assessed the combination of sipuleucel-T and a radioisotope, radium-223, in patients with mCRPC. Despite paradoxically decreasing the immune response in the combination arm, PSA levels were decreased, and PFS and OS were longer [270]. A case report by Han et al. presents a significant clinical response to the pembrolizumab and radiation combination in a patient heavily treated for mCRPC with rectal involvement. After radiation and six cycles of the drug, the PSA was undetectable, the prostate mass was decreased, and the rectal invasion was imperceptible in imaging studies [271].

## 7. Overview of the Immunomodulatory Effects of Focal Therapies and RT

Focal therapies and RT boost the immune response in multiple ways. As previously mentioned, it is difficult to make an accurate comparison of the immunomodulatory effects due to miscellaneous reasons, including the complexity of the immune system, a variety of possible immunological outcomes, and lack of clinical trials assessing the differences between therapies. Moreover, it is worth mentioning that most studies do not distinguish an additive effect of a therapy from a synergistic effect, which concerns studies on cytokines expression, in particular, but there are articles that correctly incorporate both terms [272,273,274,275,276]. In the case of studying the immunomodulatory effects of focal therapies, an additive effect may be defined if the level or value of a certain immune property is either equal or lower in the combinational approach than the sum of the values of the effect in the monotherapeutic approach (1 + 1 ≤ 2). A synergistic effect occurs when the level of analyzed immune effect is higher in the combinational approach than the sum of the values in monotherapies alone (1 + 1 ≥ 2) [277,278]. This nuance may have clinical implications in future studies and should be carefully analyzed in multi-arm clinical trials assessing the immune effect of RT and focal therapies.

Table 2 presents a thorough overview of the immunomodulatory impact of different local treatment strategies on the TME.

## 8. Conclusions

Immunotherapy for PC has been thoroughly explored in recent years. Despite the initial success of sipuleucel-T, it seems that immunotherapy shows lower efficacy in PC in comparison to other malignancies. Further phase I/II clinical trials investigating radioimmunotherapy and combinations of focal and immune therapies are highly desirable. The first studies concerning the immunomodulatory properties of RT and focal therapies cautiously showed promising results, although a well-planned randomized clinical trial, addressing both immunological and clinical outcomes, would be necessary to accurately assess the usefulness of immunotherapy and its combinations in PC management. A meticulously designed study such as this could plausibly prove the synergistic effect provided by the combination of therapies, which would hopefully create new clinical perspectives for focal treatment options.

## Figures and Tables

**Figure 1 cells-11-02793-f001:**
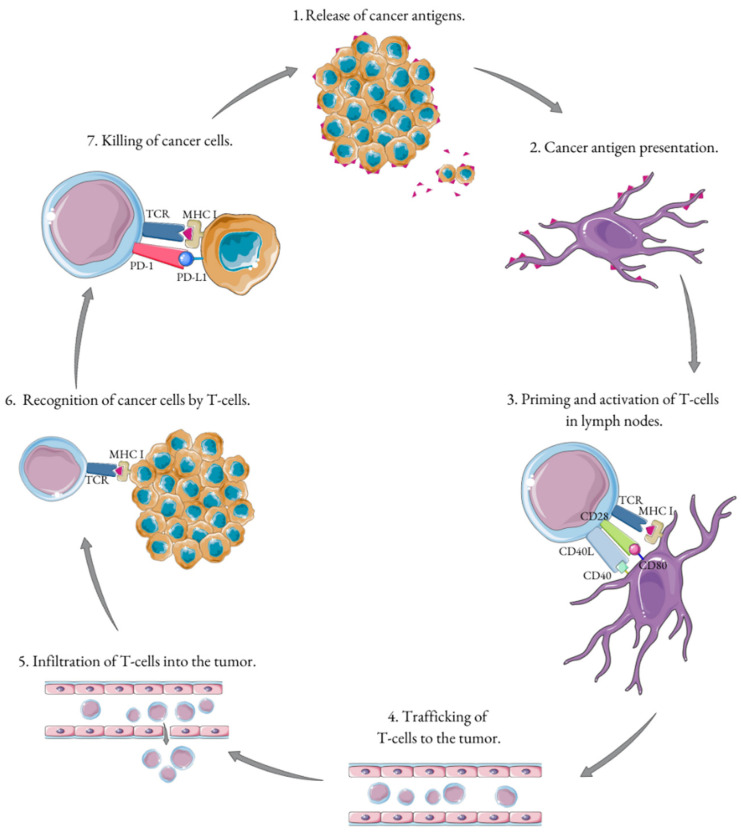
Schematic overview of cancer immunity cycle.

**Table 1 cells-11-02793-t001:** Ongoing trials assessing combinations of radiotherapy and immunotherapy.

NCT Number	Phase	n	Setting	Immunotherapeutics	Radiotherapy
NCT03835533	I	45	mCRPC	NKTR-214, Nivolumab, CDX-301, Poly-ICLC, INO-5151	SBRT
NCT03795207	II	96	mPC	Durvalumab	SBRT
NCT03543189	I/II	44	PC	Nivolumab	Brachytherapy, EBRT
NCT03217747	I/II	173	mCRPC	Anti-OX40, Avelumab, Utomilumab	RT *
NCT03007732	II	42	PC	Pembrolizumab, SD-101	SBRT
NCT01818986	II	20	mCRPC	Sipuleucel-T	SBRT
NCT01436968	III	711	PC	Aglatimagene Besadenovec	EBRT

NCT: The National Clinical Trial; n: number of patients enrolled; PC: prostate cancer; mPC: metastatic prostate cancer; mCRPC: metastatic castration-resistant prostate cancer; RT: radiotherapy; SBRT: stereotactic body radiation therapy; EBRT: external beam radiation therapy. * The specific variant of radiation therapy was not specified.

**Table 2 cells-11-02793-t002:** Immunomodulatory impact of local treatment strategies on the TME.

Local Therapy	Immunomodulatory Effects	References
HIFU	▪ Promotion of DCs infiltration and activation. ▪ Increase in CD3+ and CD4+ levels cells, and CD4+/CD8+ ratio. ▪ Enhancement of CTLs cytotoxicity.	[187,188,189,190,191,192,193,194,195,196,197,198,199]
Cryotherapy	▪ Activation of T cells. ▪ DC maturation. ▪ The abscopal effect.	[204,207,208,209,210,211,212,213,214,215,216]
PDT	▪ Promotion of neutrophils recruitment. ▪ DC maturation. ▪ Activation of macrophages. ▪ Regulation of CTL and NK cell migration, increase in CD8+ T cell infiltration. ▪ Secretion of IL-1, IL-6, IL-8, IL-10, IL-12, thromboxane, and prostaglandins.	[133,225,230,231,232,233,234,235,236,237]
IRE	▪ Decrease in Tregs and MDSC levels.	[248,249,250,251,252]
RT	▪ Enhancement of damage-associated molecule release. ▪ Activation of DCs and other APCs. ▪ Release of various cytokines (e.g., CXC10, CXCL16, IL-1, TNF-α, and interferons). ▪ Upregulation of MHC I, NKG2D ligand and Fas/CD95. ▪ The abscopal effect.	[253,258,259,260,261,262,263,264,265,266,267,268,269,270,271]

HIFU: high-intensity focused ultrasound; PDT: photodynamic therapy; IRE: irreversible electroporation; RT: radiotherapy; DC: dendritic cell; CTL: cytotoxic T lymphocyte; NK: natural killer; IL: interleukin; Treg: regulatory T cell; MDSC: myeloid-derived suppressor cell; APC: antigen-presenting cell; TNF: tumor necrosis factor; MHC: major histocompatibility complex.

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
