# Peer review of "Boosting the Immune Response—Combining Local and Immune Therapy for Prostate Cancer Treatment"

_cells, 2022, doi:10.3390/cells11182793_

Round 1

Reviewer 1 Report

This review aimed to comprehensively summarize the advancement of local therapy and immunotherapy in prostate cancer. They introduced the background of immunology in prostate cancer, including the T cell infiltration, T cell response, and specific immune cells mediated the immunosuppressive signaling in prostate cancer. Then, they introduced the prostate cancer vaccines in prostate cancer. Finally, they comprehensively summarized the focal/local therapy in combination with immunotherapy in prostate cancer.  Overall, this review is straightforward and important. Here are a couple of points shared by the authors:

1.     The structure of the review need be adjusted. It is unclear the connection of the five sections in this review. From immunology background jumps to cancer vaccines and then to combination therapies.

2.     The important immune checkpoint inhibitors were only mentioned a small paragraph in section 2.5 which is merged in immunology background section. For the background, the authors need introduce the current immune checkpoints in prostate cancer.

3.     The cancer vaccines seem out of the context in this review. It did not show any information of focal/local therapy in combination with vaccines in this section.

4.     Section 4 and 5 are important. However, section 5 needs to add sub-titles to cover different angles.

Reviewer 2 Report

Dear Authors,

The manuscript focus on focal therapy for prostate cancer, its possible immunological effects and the possible benefit in combination with immunotherapy. As such I think the topic is of interest and it is some aspects well written. I agree with the scope and the way they handled the topic in general. However, there is some issue needed to be addressed for clarity. Over all I suggest that the authors in the beginning need to stipulate what the mean with immune properties of focal therapy. What ascpects of immune response do they mean. What aspects do they not address? How did they do the selection of articles? Further the manuscript need to have some kind of summary of what kind of response could be seen with the different therapies and how to monitor them, is there a gold standard of immunological effect. In the article it is a little confusing and cathegorization would perhaps be added.  Further the article discuss the possibility that immunotherapy and focal therapy have a synergistic effect, it would be of interest if they could stipulate the scientific base for two drugs to give a synergistic effect (1+1=3) and not only two additative effects (1+1=2). Most studies performed as I have seen is investigating the additative effect not the synergistic effect. 

In the abstract the author state tha focal therapy is restricted to localized and low-grade prostate cancer (PC), as if it is accepted as standard of care in this situation. I have not knowledge of all guidelines, but EAU guidelines state …“the available evidence indicate that focal therapy should be performed within the context of a clinical trial setting or well-designed prospective cohort studies.” Please rephrase the sentence in the abstract.

At row 44 first page the author state 

“..promising oncological results, especially from a short-term perspective…” I would like the authors to be more clear what they mean with this sentence. As I read the refeered article about focal therapy there is a potential risk of recurrence of HIFU 

…The in-field recurrence rate was up to 40% after 12 months according to von Hardenberg (Urol Oncol2018; 36:401.e1–401.e9.)

and the risk of metastases within 5 years seem rather higher in comparison to the PROTECT trials 10 years follow up.

… And Guillaumier S et al (Eur Urol 2018; 74:422–421.) show mean 5 year metastatic free survival in low risk patients of 95% in HIFU patients ... The PROTECT trial there were only 6% of the patient in active surveillance that had metastases after 10 years follow up and about 2% in the surgery group …

At row 50 they state “clinical benefit of immunotherapy alone  remains limited due to low-grade inflammation in the tumor microenvironment (TME) “ is there enough evidence for this statement or is it the view of the authors, please rephrase and add either the evidence for this statement or rephrase it as a statement that is based on the authors opinion.

In the section of the immune escape mechanism, I have thought that the hypoxia in the tumors also affect the immune response in the tumors, is that hypothesis falsified?

Are there any difference between the immune escape mechanisms in the primary tumor microenvironment and the metastatic sites? Even if there is or if the authors do not think there is I would be very interested in hearing the authors discussion about that topic, and also if there is not known in the literature I would suggest that it could be mentioned anyhow.

Of notice: The RhoVac phase II study was recently announced to be negative (https://www.rhovac.com/cision/777FBD7CCEADBD45/)

I would suggest that the authors state under each section of RT, HIFU…if there is any clinical data in prostate cancer on the primary objective ot the review, “immune properties” ( I read immunological response to) the treatment modalities, if not please state that. And please state what kind of immune response that is triggered by the different focal therapies.

I would also be interested to read a paragraph of what end point or surrogate end point to measure the “immune properties” of any modality. What is the gold standard to see an immunological effect of a treatment? Is the absobal effect the gold standard, if so does it need to show complete remission or does regression be enough? Or is immune cell infiltration the best way to see an effect? I would be very happy to read your thoughts about this. 

Row 49-50.. I disagree with the statement the authors use here, it sounds like it is proven and generally accepted that the only reason for lack of benefit with immunotherapy is due to low-grade inflammation in the tumor microenvironment. Please provide evidence for that statement or change it to a more objective phrasing.

Row 137 I do not think MDSC is previously abbreviated, please add the explaination.

Row 172 you describe other mechanism of TGF-beta and I suggest that “thus” is changed to “and” or have I misunderstood anything?

Row 181 you describe the MHC presentation and I suggest that you add reference to the work from Pernilla wikströms group (Ume University) showing that there is a specific molecular subgroup of prostate cancer metastases that have less expression of MHC I expression, who suggest that some molecular sub-groups may be less responsive to immunotherapy. 

Row 216-228 the authors discus a study with vaccination and docetaxel.. It is a interesting study but I do not see that it is in the scope of the review with focal therapy and immune properties.

At row 247 you state that the combination of enzalutamide and vaccination did not affect the PSA level… The Effect of antiandrogens and immune response is of great interest but does in my opinoion either be outside the scope of this article or should be mentioned in a specific paragraph of focal therapy with or without antiandrogens? For example: (Asim, M., Tarish, F., Zecchini, H.I. et al. Synthetic lethality between androgen receptor signalling and the PARP pathway in prostate cancer. Nat Commun 8, 374 (2017). https://doi.org/10.1038/s41467-017-00393-y)

Row 271 you have a sentence that either is redundant or needs more explanation why it has something to do with the review at hand.

At row 341-344 the authors state that the effect of cryotherapy and its “immunosuppressive impact” differ in the peripheral and the center area of cryotherapy, is this by evidence shown or is it and suggested hypothesis/opinion of the authors? I suggest another way of phrasing this suggested difference. And if there is such a difference, does it impact how we should think of treatment with this modality?

At row 356-357 the authors once again is very confident in the concept of synergistic effect of immunotherapy and focal therapy. I disagree with the use of synergistic effect in this statement. There is no possibility to discuss a synergistic effect in a prospective non-randomized trial. The word synergistic does to me state that one plus one is three and the study is not performed in a way that this could be assessed. (perhaps add in the conclusion that studies are needed to investigate the synergistic effect of focal therapy and immunetherpay, i.e. if Lin et all should have had three arms (NK cell only, Focal therapy only or the combination a synergistic effect could have been investigated).

In the paragraph of 370 or in the next row I think you need to print out photodynamic therpay(PDT) for clarity if not done previously in the manuscript.

In the whole paragraph of PDT you seem very positive about the therapy and it does seem to be influence by a opninion of the athours to be a very good treatment and at row 390 you describe and seem to critize the FDA committee decision in a very subjective way. I do not agree with the way the authors seem to be writing in a non-objective manor and suggest a rephrasing the paragraph to some degree for scientific stringency.

At row 409-411 I was a little confused… are the authors refering to a specific PDT therapy or is this therapy some other sort of therapy? Please specify.

At row 420 “demonstrates vaccine-like function..” please specify what you mean, does not all discussion about immune properties of focal therapy include a vaccine-like function..?!? Antibody production or T-cells response or what do you mean? once again the manuscript would increase in clarity if you initially stipulate what you mean with the immune properties that you refere to as the main objective of the study.

At row 464 you state that the immune-stimulating effect of RT is generally achieved through induced cell death and modulating the composition of TME. The first statement seem clear but the second is somewhat elusive, what do you mean and is there evidence of that statement or is it a hypothesis/opinion?

I also disagree with all the statements in the conclusion:

Is really immunotherapy an unexplored field? Is sipuleucel-T a success or did it not be as good as it hope to be? Is it not more accurate to say that immunotherapy has not been as successful in prostate cancer as in other tumor. 

Furhter I disagree with the next two sentences “the RT and immunotherapy combo is an approach of the greatest potential to increase anti-tumor qualities of TME. Thus, it may be the most effective strategy stimulating the cancer-related immune response in PC.”

In my opinion the review gives any clear evidence for such a statement. The only way to say that RT is better in combination with immunotherapy would be to have a randomized trial between immunotherapy with RT versus any other treatment and In my opninon there should need to be a control arm with only immunotherapy alone. I would suggest another conclusion stating the need for well planned and conducted clinical trial, of different types. For example with different end points addressing the difference in immune response that could be possible based on the immune therapy and focal treatment respectively.

Round 2

Reviewer 1 Report

The authors addressed the concerns

Reviewer 2 Report

Dear Authors,

The manuscript is improved and there are only a few places needed to be adressed.

row 206 and 217 there is a lack of space for the paragraph.

row 630-531, is it really only studies on cytokine expression that have poor distinguishing between additive and synergistic effect???

row 634 I guess you could write 1+1≤2 and row 636 it would b ≥2
